# *C. elegans* Germline as Three Distinct Tumor Models

**DOI:** 10.3390/biology13060425

**Published:** 2024-06-08

**Authors:** Mariah Jones, Mina Norman, Alex Minh Tiet, Jiwoo Lee, Myon Hee Lee

**Affiliations:** 1Division of Hematology/Oncology, Department of Internal Medicine, Brody School of Medicine at East Carolina University, Greenville, NC 27834, USA; jonesmari20@students.ecu.edu (M.J.); normanm18@students.ecu.edu (M.N.); 2Neuroscience Program, East Carolina University, Greenville, NC 27858, USA; tieta20@students.ecu.edu; 3Department of Biology, East Carolina University, Greenville, NC 27858, USA

**Keywords:** tumorigenesis, GLP-1/Notch signaling, RNA-binding proteins, GLD-1, PUF-8, *C. elegans* germline

## Abstract

**Simple Summary:**

Both extrinsic signaling and intrinsic regulation are critical for maintaining cellular homeostasis, and their dysregulation is often associated with tumorigenesis and human diseases. This report outlines three distinct *C. elegans* tumor models resulting from mutations in conserved extrinsic signaling pathways (e.g., Notch signaling) and intrinsic RNA-binding proteins (e.g., GLDs and PUF). These models highlight how *C. elegans* Notch signaling and RNA-binding proteins contribute to tumor initiation, progression, and suppression, depending on the cellular context. Therefore, in addition to targeting oncogenic signaling pathways, directing attention toward RNA-binding proteins holds great potential for a tumor-type-specific therapy approach.

**Abstract:**

Tumor cells display abnormal growth and division, avoiding the natural process of cell death. These cells can be benign (non-cancerous growth) or malignant (cancerous growth). Over the past few decades, numerous in vitro or in vivo tumor models have been employed to understand the molecular mechanisms associated with tumorigenesis in diverse regards. However, our comprehension of how non-tumor cells transform into tumor cells at molecular and cellular levels remains incomplete. The nematode *C. elegans* has emerged as an excellent model organism for exploring various phenomena, including tumorigenesis. Although *C. elegans* does not naturally develop cancer, it serves as a valuable platform for identifying oncogenes and the underlying mechanisms within a live organism. In this review, we describe three distinct germline tumor models in *C. elegans*, highlighting their associated mechanisms and related regulators: (1) ectopic proliferation due to aberrant activation of GLP-1/Notch signaling, (2) meiotic entry failure resulting from the loss of GLD-1/STAR RNA-binding protein, (3) spermatogenic dedifferentiation caused by the loss of PUF-8/PUF RNA-binding protein. Each model requires the mutations of specific genes (*glp-1*, *gld-1*, and *puf-8*) and operates through distinct molecular mechanisms. Despite these differences in the origins of tumorigenesis, the internal regulatory networks within each tumor model display shared features. Given the conservation of many of the regulators implicated in *C. elegans* tumorigenesis, it is proposed that these unique models hold significant potential for enhancing our comprehension of the broader control mechanisms governing tumorigenesis.

## 1. Introduction

Tumorigenesis, the abnormal proliferation of cells leading to tumor formation, highlights various capabilities of cancer cells, such as growth signal self-sufficiency, insensitivity to anti-growth signals, apoptosis evasion, replicative potential, sustained angiogenesis, and tissue invasion [1]. Due to the complexity and overlapping genetic changes in humans, model organisms are crucial in studying tumorigenesis. The nematode *Caenorhabditis elegans* (*C. elegans*) is a good model organism for studying fundamental mechanisms of cell proliferation and differentiation and is one possible model for studying tumorigenesis in vivo. The transparency of *C. elegans* enables direct observation of cell growth and tumorigenesis in vivo [2]. Its highly conserved genes and pathways related to tumorigenesis make *C. elegans* an ideal model organism [2], especially for studying tumorigenesis in germlines influenced by both germline and somatic signals [2].

### 1.1. C. elegans Germline Development

*C. elegans* exist as either hermaphrodites (XX) or males (XO). Hermaphrodites initially produce a limited number of sperm during the larval stage (L4) and then switch to producing oocytes in young adult stages; they are self-fertile during adulthood. However, males consistently produce sperm without switching to oogenesis. The germline is organized in a simple linear pattern in both sexes, progressing from germline stem cells (GSCs) in the distal region to maturing gametes in the proximal region (Figure 1A). Specifically, a mesenchymal somatic cell, known as the distal tip cell (DTC), functions as a GSC niche and plays a crucial role in GSC maintenance and the mitotic cell cycle in the distal germline [3]. Once a GSC moves away from the DTC niche, it enters the meiotic cell cycle and eventually differentiates into sperm or oocytes (Figure 1A). In addition, *C. elegans* is an attractive model organism due to the ease of generating mutant strains, genetic manipulation, phenotype analysis, microscopy, and imaging. Therefore, the *C. elegans* germline has been widely used as a model organism in various biomedical fields, including research on tumorigenesis.

### 1.2. Three Distinct Mechanisms of C. elegans Germline Tumorigenesis

Germline tumors observed in mutant *C. elegans* individuals can originate from different sources, depending on the mechanism of tumorigenesis. In the context of *C. elegans*, the term “tumorous” is used to define germlines that exhibit three distinctive features (Figure 1B): (1) a vast excess of mitotic cells through ectopic proliferation, (2) minimal or no germ cell differentiation due to meiotic entry failure, and (3) the generation of mitotic germ cells through dedifferentiation. This report outlines three distinct mechanisms of tumorigenesis.

## 2. GLP-1/Notch-Activation-Mediated Tumorigenesis: Ectopic Proliferation

### 2.1. Notch Signaling

In most multicellular organisms, the Notch signaling pathway is highly conserved and controls various cellular processes, including proliferation, differentiation, cell fate specification, and other cellular responses [4]. Notch ligands (DSL: Delta/Serrate/LAG-2) are expressed on the membrane of donor cells adjacent to receiving cells expressing Notch receptors (Figure 2A). Upon interaction between the Notch receptor and a ligand, an ADAM-family metalloprotease cleaves the exterior of the Notch receptor, followed by γ-secretase cleaving the inner portion of the Notch receptor within the cell membrane. This Notch intracellular domain (NICD) relocates to the nucleus, where it forms a tertiary complex with CSL (CBF1/Suppressor of Hairless/LAG-1) transcription factors and Mastermind-like protein (MAML-1), leading to the activation of Notch target genes. Notably, aberrant Notch signaling can lead to uncontrolled cell growth, metastasis, and resistance to apoptosis [5], which are associated with breast cancer, lung adenocarcinoma, hepatocellular cancer, ovarian cancer, and colorectal cancer [5]. Therefore, understanding the precise mechanism governing context-dependent outcomes of Notch signaling is crucial.

### 2.2. C. elegans Notch Signaling and Its Core Regulators 

The Notch signaling pathway and its core regulators are highly conserved in *C. elegans* (Figure 2B). Two distinct Notch receptors exist in this organism: GLP-1 (GermLine Proliferation-1) and LIN-12 (cell LINeage-12). GLP-1 is primarily found in germ cells and plays a critical role in the maintenance and mitotic division of GSCs [3]. Conversely, LIN-12 is predominantly present in somatic cells and is essential for determining the fate of vulva cells during early larval development [6]. The activation of GLP-1 signaling requires two Notch ligands: LAG-2 and APX-1 [7,8]. LAG-2 is mainly found in the DTC, which acts as a niche for GSCs. When LAG-2 interacts with GLP-1, a cascade of cleavage events is initiated, leading to the release of the NICD [9]. Subsequently, GLP-1(NICD) forms a tertiary complex with LAG-1 and LAG-3, activating transcription for its target genes [9]. Two target genes, lateral signaling target (*lst-1*) and synthetic GLP (*sygl-1*), have been extensively studied [10]. These genes function redundantly in the maintenance and mitotic division of GSCs [10]. While a single mutation for either *sygl-1* or *lst-1* can sustain GSCs, double mutations show no GSC phenotype, similar to *glp-1*(*null*) mutants [10]. Notably, overexpression of either *sygl-1* or *lst-1* induces the formation of germline tumors, resembling the phenotype observed in *glp-1* gain-of-function (gf) mutants [11]. These findings highlight the crucial role of two GLP-1 target genes in the maintenance and mitotic division of GSCs.

### 2.3. GLP-1 Mutant Alleles

The primary distinction between the two groups of mutants lies in their germline phenotype. Loss-of-function mutants significantly reduce GSCs, while gain-of-function mutations form germline tumors. These *glp-1* mutants have served as valuable tools to identify genes associated with GLP-1 signaling. The representative mutant alleles are listed in Table 1.

### 2.4. Positive or Negative Regulators of GLP-1/Notch Signaling

While most *glp-1(ar202)* and *glp-1(bn18)* mutants are fertile at 15 °C or 20 °C, the majority of *glp-1(ar202)* gain-of-function mutants develop germline tumors upon shifting to 25 °C [15], whereas most *glp-1(bn18)* loss-of-function mutants exhibit defects in germline proliferation at 25 °C, resulting in no germ cells [12]. These intriguing phenotypes provide an opportunity to identify genes that positively or negatively regulate GLP-1/Notch signaling (Figure 2C).

CYE-1/CDK-2 cell cycle regulators: CYE-1 and CDK-2 form a complex that plays critical roles in regulating cell cycle progression from the G1 to the S phase [16]. Fox et al. found that germlines CYE-1 and CDK-2 are required for GLP-1/Notch-mediated germ cell proliferation [16]. Specifically, a temperature-sensitive *glp-1(bn18)* loss-of-function mutant can maintain proliferative germ cells at 15–20 °C, but it loses them at 25 °C [12]. Notably, RNAi-mediated depletion of CYE-1 or CDK-2 significantly suppressed germ cell proliferation in these mutants even at 20 °C [16]. Additional genetic analysis suggests that CYE-1 and CDK-2 act independently of GLP-1/Notch signaling to promote germ cell proliferation [16].Subunits of the DNA polymerase alpha–primase complex: Yoon et al. found that DIV-1 (regulatory subunit) is indispensable for GLP-1/Notch-mediated germ cell proliferation during early larval development, whereas POLA-1 (catalytic subunit) and two primase subunits, PRI-1 and PRI-2, play a crucial role in GLP-1/Notch-mediated maintenance of proliferative cell fate during adulthood [17]. Robinson-Thiewes et al. also identified POLE-1 (the catalytic subunit of DNA polymerase e) as a regulator of germ cell proliferation [18].Chaperone HSP90: Lissemore et al. performed genetic screening to identify genes that promote GLP-1/Notch signaling and found that HSP-90, a molecular chaperone, plays an essential role in stem cell maintenance [19]. It was a novel finding demonstrating the essential role of HSP90 in Notch signaling in development.Ribosomal protein S6 kinase (S6K): Roy et al. identified RSKS-1/S6K as a positive regulator of GLP-1/Notch-signaling-mediated germline proliferation [20]. Additional screening also found that translation-related proteins, *cacn-1*/Cactin, an RNA exosome component, and a Hedgehog-related ligand may share functional relationships with GLP-1/Notch and RSKS-1/S6K in maintaining GSCs [20].Bro1-domain protein: Liu and Maine identified the *ego-2* (enhancer of *glp-1*) gene as a positive regulator of germline proliferation that interacts genetically with the GLP-1/Notch signaling pathway [21]. Notably, *ego-2* also promotes LIN-12/Notch signaling in somatic tissues [21]. They found that the EGO-1 protein contains a Bro1 domain, which localizes to specific endosomal compartments in other systems. Thus, they suggest that EGO-2 may promote GLP-1/Notch signaling through am endocytic process function [21].Derlin family proteins: Singh et al. demonstrated that reduced CUP-2 and DER-2 function suppresses GLP-1/Notch-mediated germline tumorigenesis [22]. CUP-2 and DER-2 are Derlin family proteins that function in endoplasmic reticulum (ER)-associated degradation (ERAD). They also found that the suppression of GLP-1/Notch-mediated germline tumorigenesis by the *cup-2* mutation requires a proper Unfolded Protein Response (UPR) function. Therefore, they suggest that reduced Derlin activity may suppress GLP-1/Notch-mediated tumorigenesis through the activation of ER stress and UPR [22].U/T level: Chi et al. demonstrated that *C. elegans* CDD-1/-2 cytidine deaminases are involved in uridine biosynthesis [23]. Notably, worms lacking both CDD-1 and CDD-2 exhibited germline proliferation defects, whose phenotype was rescued by uridine/thymidine (U/T) supplementation [23]. They also suggested that U/T levels regulate the translation of *glp-1* mRNA through its 3′UTR in the distal mitotic region at the post-transcriptional level [23].TRIM-NHL protein: Brenner et al. identified *nhl-2* as an inhibitor of *glp-1(ar202)*-mediated tumorigenesis [24]. NHL-2, a conserved TRIM-NHL protein family member, suppresses germ cell proliferation by inhibiting two PUF RNA-binding proteins, PUF-3 and PUF-11 [24]. They also found that CGH-1 RNA helicase and ALG-5 miRNA-associated Argonaute work with NHL-2 to inhibit *glp-1(ar202)*-mediated tumorigenesis [24].E3 Ubiquitin ligase: Gutnik et al. reported that the splicing factor PRP-19 (a candidate E3 ubiquitin ligase) inhibits the nuclear accumulation of the GLP-1/Notch intracellular domain [25].PUF RNA-binding protein: PUF-8 is a conserved PUF RNA-binding protein that inhibits the translation of target mRNAs [26,27]. In *C. elegans* germline, PUF-8 is involved in decisions regarding proliferation/differentiation, differentiation/dedifferentiation, and sperm/oocyte fates, depending on the genetic context [28]. Racher and Hansen demonstrated that PUF-8 inhibits *glp-1(ar202)*-mediated tumorigenesis in the *C. elegans* germline [29]. Other PUF RNA-binding proteins (FBF-1/2 and PUF-3/11) act downstream of GLP-1/Notch signaling and play a critical role in GLP-1/Notch-signaling-mediated germ cell proliferation and tumorigenesis [30].Syndecan: Gopal et al. identified SDN-1 (a syndecan transmembrane proteoglycan) as a positive regulator of GLP-1/Notch signaling. SDN-1 promotes GLP-1 expression and mitotic germ cell fate by controlling a somatic TRP calcium channel [31]. This TRP channel enhances *glp-1* expression by governing the calcium-dependent binding of the APTF-2 transcription factor [31]. Notably, the *glp-1* promoter has an APTF-2 binding site, and its transcription is directly activated by APTF-2 [31].

## 3. GLD-1-Loss-Mediated Tumorigenesis: Meiotic Entry Failure

### 3.1. STAR Family of RNA-Binding Proteins

The signal transduction activator of RNA metabolism (STAR) family of KH domain RNA-binding proteins is a highly conserved group of proteins among eukaryotes [32]. The STAR protein family operates at the post-transcriptional level, involved in the stability, alternative splicing, and translational efficiency of their mRNA targets, thus influencing downstream gene expression [33]. Although STAR proteins are highly conserved, they exhibit variability in specific regions and specificity to certain RNA sequences in each model organism [34]. Moreover, RNA-recognition mechanisms by STAR proteins are yet to be explicitly defined for many models. Many STAR proteins have been identified with diverse roles across animal models [35]. For example, the held-out wing (HOW) gene in *Drosophila* is involved in embryonic cardiac development [36] and germline differentiation [37], and the alternative splicing defective-2 (ASD-2) gene in *C. elegans* is involved in the developmental control of alternative splicing [38,39]. STAR proteins also play an essential role in the development of vertebrate models. Most notably, mutations and loss of function in the quaking gene (QK1) in mice models have been shown to cause deficiencies in adult mice astrocyte maturation [40]. Abnormal STAR proteins have also been implicated in tumorigenesis, including lung cancer [41,42], breast cancer [43], and colorectal cancer [44]. Therefore, more research is needed to elucidate the full pleiotropic effects and mechanisms of STAR proteins and their role in developmental maturation and tumorigenesis.

### 3.2. C. elegans gld-1 and Its Partners

*C. elegans* STAR RNA-binding protein GLD-1 (GermLine development Defective) plays multiple critical roles in *C. elegans* germline development. One well-known function is inhibiting germ cell proliferation [45,46]. Germ cells lacking GLD-1 enter the meiotic cell cycle and revert to the mitotic cell cycle, leading to the formation of germline tumors (Figure 1B). GLD-1 is predominantly expressed in the cytoplasm of premeiotic and pachytene cells in the *C. elegans* germline [46]. GLD-1 exerts its regulatory functions by binding to conserved sequence motifs (AGAAGC, CUACUAAC, or GAACGA) in the 5′ and 3′ UTRs of its mRNA target [47,48] (Figure 3A), thereby modulating their stability and/or translation [49]. Several GLD-1 target mRNAs have been identified through biochemical and functional analyses, including *rme-2*, *gna-2* [50], *lin-45*, *tra-2*, *glp-1* [51], *pos-1*, *pal-1* [52], *cye-1*, *cep-1* [53], *mes-3* [54], and five *puf* genes (*puf-5*, *6*, *7*, 8, and *10*) [48,55]. RNA-IP/Chip analysis has further identified putative GLD-1 targets, primarily involved in reproduction, embryogenesis, cell division, and the cell cycle [49]. GLD-1 and its partner NOS-3 (a member of the Nanos family of zinc finger proteins) act as a translational repressor to promote meiotic prophase progression in the *C. elegans* germline [9,56]. Moreover, the GLD-1/NOS-3 complex works with the GLD-2/GLD-3 complex, which promotes germline differentiation [9,56]. Thus, the GLD-1/NOS-3 and GLD-2/GLD-3 complexes are essential for germline differentiation, particularly through meiotic entry (Figure 3B).

### 3.3. gld-1 Mutant Alleles

The *gld-1* gene plays a crucial role in meiotic progression and oocyte development [45]. The formation of germline tumors in *gld-1(q485)* loss-of-function mutants is dependent on the sexual fate of the germline [58]. In the *gld-1(q485)* loss-of-function mutant germline, germ cells exit the meiotic prophase but return to the mitotic cell cycle in the oogenic germline, while these germ cells can differentiate into sperm in the spermatogenic germline [58]. The representative mutant alleles are listed in Table 2.

### 3.4. Positive or Negative Regulators of GLD-1

The regulators of GLD-1 are depicted in Figure 3B.

GLD-2 poly(A) polymerase (PAP): GLD-2 is a cytoplasmic poly(A) polymerase [60]. It plays a critical role in meiotic entry and progression [61,62]. Thus, no functional gametes are produced in the absence of GLD-2 [61]. Notably, the *gld-1* mRNA is a direct target of GLD-2 [63]. GLD-2 promotes meiotic entry at least in part by activating the translation of *gld-1* mRNAs [63]. Consequently, GLD-2 loss enhances the formation of germline tumors in *gld-1* loss-of-function mutant worms [61].FBF/PUF RNA-binding protein: *C. elegans* FBF/PUF proteins play a crucial role in maintaining GSCs by regulating the expression of various target mRNAs, including the *gld-1* mRNA [64]. Since GLD-1 is essential for inhibiting proliferation and maintaining the differentiation state of germ cells, FBF/PUF repression of *gld-1* mRNAs is critical for GSC maintenance. In addition, *C. elegans* PUF-8 proteins negatively regulate the abundance of GLD-1 proteins via the inhibition of *gld-2* mRNA translation [27].CYE-1/CDK2: GLD-1 has CDK2 phosphorylation sites and appears to be a direct substrate of CYE-1/CDK2 [65]. Functional analysis showed that FBF and CYE-1/CDK2 maintain GSCs by inhibiting GLD-1 abundance in the distal mitotic region through post-transcriptional and post-translational mechanisms, respectively. Moreover, *cye-1* mRNA is also a repressing target of GLD-1 [66]. Therefore, GLD-1 and CYE-1/CDK2 inhibit each other for the mitosis/meiosis balance (Figure 3B).Pre-mRNA splicing factor (PRP-17): Kerins et al. reported that PRP-17 and other *C. elegans* splicing factor orthologs function to promote meiotic entry by positively regulating the splicing of mRNAs of genes in the GLD-1 pathway [67].

## 4. PUF-8-Loss-Mediated Tumorigenesis: Spermatogenic Dedifferentiation

### 4.1. PUF RNA-Binding Proteins

Pumilio and FBF (PUF) proteins are highly conserved stem cell regulators that maintain GSCs in worms and flies and have also been identified in vertebrate stem cells [28,68,69,70,71,72]. These proteins control mRNA translation or stability by binding to regulatory elements in the 3′ UTR of their target mRNAs (Figure 4A). Specifically, they repress the expression of target mRNAs by recruiting the Ccr4-Pop2-NOT deadenylase complex to trim the poly(A) tails [73] and/or by interacting with Argonaute proteins to stall translation elongation [74] (Figure 4A). Notably, many PUF-repressing target mRNAs repressed by PUF proteins are shared among worms, flies, and humans, including components of cell signaling, cell cycle regulation, and development [75]. One of the conserved target mRNAs is an ERK MAPK mRNA [71]. PUF proteins inhibit the expression of MAPK mRNAs in both *C. elegans* and human embryonic stem cells [71]. In humans, two PUF proteins, PUM1 and PUM2, have distinct roles in the self-renewal and differentiation of mesenchymal stem cells (MSCs) [72]. PUM1 is critical for MSC self-renewal and proliferation, while PUM2 represses the osteogenic differentiation of MSCs by inhibiting *JAK2* and *RUNX2* mRNAs [72]. Since cancer stem cells have similar characteristics, several studies have highlighted the novel function of PUM1 in cancer stem cells and cancer progression [76,77,78]. For example, Pumilio proteins promote colorectal cancer progression by inhibiting the expression of p21 mRNA [79]. Therefore, PUF proteins play critical roles in regulating various cellular processes and tumorigenesis at a post-transcriptional level.

### 4.2. C. elegans PUF-8

*C. elegans* possesses 11 PUF genes [80] (Figure 4B). Among them, PUF-8 is predominantly expressed in the distal germline and plays a key role in regulating various cellular processes, depending on the genetic context in the *C. elegans* germline [28]. For example, PUF-8 and MEX-3 are critical for GSC proliferation [81], whereas PUF-8 and LIP-1 promote GSC differentiation and the oogenic fate by inhibiting the MPK-1/ERK signaling pathway [82]. PUF-8 also works with FBF-1 to promote the oogenic fate and inhibit the spermatogenic fate [83]. The PUF-8 protein recognizes a regulatory element [UGUAnA(U/A)A] on the 3′UTR of target mRNAs [27]. In silico analysis has identified approximately 800 genes harboring at least one PBE in their 3′UTRs [27]. mRNA-seq analysis revealed that 4638 genes were upregulated and 4855 genes were downregulated in the *puf-8(ok302)* null mutant [84]. Notably, about 500 genes contained the PBE sequence in both groups of genes [84]. To date, biochemical analyses, such as yeast-three hybrid, gel shift, or RT-PCR, have verified several targets of PUF-8 regulation, including *gld-2* [27], *ced-3* [85], *let-60* [26], *pqm-1* [84], *pha-4* [84], *blmp-1* [84], *vhp-1* [84], and *hih-30* [84]. Recent reports also indicate that PUF-8 proteins regulate the *C. elegans* lifespan through pathways involving MFF (mitochondria fission factor) and *pqm-1*-related lipid storage [84].

### 4.3. puf-8 Mutant Alleles

*puf-8* loss-of-function mutants exhibit fertility at a permissive temperature (20 °C) but develop partial germline tumors via dedifferentiation at a restrictive temperature (25 °C) [86,87]. Specifically, abnormal spermatocytes in *puf-8* loss-of-function or null mutant germlines return to the mitotic cell cycle via dedifferentiation mechanisms [86,87]. Additionally, as noted in Section 2.4, *puf-8* loss-of-function or null mutations can induce germline tumors in temperature-sensitive *glp-1(ar202)* gain-of-function mutant germlines even at a permissive temperature (20 °C) through non-dedifferentiation-mediated mechanisms. The representative mutant alleles are listed in Table 3.

### 4.4. Positive or Negative Regulators of PUF-8

The regulators of PUF-8 are depicted in Figure 4B.

LIP-1 dual-specificity phosphatase: *puf-8(q725)* mutants are self-fertile at 20 °C. However, at 25 °C, ~10% of 1-day adult *puf-8(q725)* mutants develop germline tumors [87]. Notably, the germline tumor phenotype of *puf-8(q725)* mutants is dramatically enhanced by the additional loss of LIP-1 (an MPK-1/ERK inhibitor) [87]. This finding indicates that PUF-8 works with LIP-1 to inhibit dedifferentiation-mediated tumorigenesis by promoting the meiotic division of spermatocytes in the *C. elegans* germline [87].MPK-1/ERK MAPK: The Ras-ERK/MAP kinase signaling pathway governs many cellular processes, such as proliferation, differentiation, cell fate decision, and survival in most eukaryotes [88]. Components of the *C. elegans* Ras-ERK pathway, such as LET-60/Ras and MPK-1/ERK, are highly conserved and essential for germline development, including meiotic progression, sperm fate specification, and oocyte maturation [89]. Notably, the reduction in Ras-ERK MAPK signaling, either by mutation or chemical inhibition, blocked the initiation of dedifferentiation in *puf-8(q725)*; *lip-1(zh15)* mutant germlines [87,90]. These findings indicate that MPK-1/ERK signaling pathways are critical for *puf-8(q725)* dedifferentiation-mediated tumorigenesis.GLD-1 and GLD-2: Park et al. recently reported that PUF-8 binds specifically to a PBE in *gld-2* 3′UTR and represses a GFP reporter gene carrying *gld-2* 3′UTR in the *C. elegans* mitotic germ cells [27]. Notably, the removal of both *gld-2* and its activating target, *gld-1*, significantly increased *puf-8(q725)* dedifferentiation-mediated germline tumors [27]. These results indicate that GLD-1 and GLD-2 may inhibit dedifferentiation-mediated germline tumors in a *puf-8(q725)* mutant germline by promoting germ cell differentiation.

## 5. Conclusions

Tumorigenesis can occur through various mechanisms, such as genetic mutations, epigenetic alterations, chromosomal abnormalities, immune system dysfunction, changes in the tumor microenvironments, and metabolic changes. In this report, we describe three specific tumor models: tumorigenesis mediated by (1) aberrant GLP-1/Notch activation, (2) *gld-1* loss-of-function-mediated differentiation failure, and (3) *puf-8* loss-of-function-mediated dedifferentiation (Figure 1B). Notably, each model requires the mutation of distinct genes (*glp-1*, *gld-1*, and *puf-8*) and operates through different underlying mechanisms. However, regulators within each tumor model exhibit shared features. For example, a *puf-8(q725)* mutation initially triggers dedifferentiation-mediated tumorigenesis, further enhanced by the additional mutation of *lip-1*. Interestingly, the same *puf-8(q725)* mutation also induces *glp-1(ar202)*-mediated tumorigenesis through a different mechanism. Similarly, a *gld-1(q485)* mutation initially leads to the formation of germline tumors due to differentiation failure, but this mutation also enhances *puf-8(q725)* dedifferentiation-mediated tumorigenesis. We also found that GLP-1/Notch signaling is involved in *puf-8(q725)* dedifferentiation-mediated tumorigenesis. These observations lead us to hypothesize that there may be shared internal regulatory networks that maintain the original tumorigenesis. This idea also suggests specific therapies targeting both the origin of tumorigenesis and the internal regulatory networks. Consequently, the three distinct tumor models in *C. elegans* serve as valuable tools for identifying genes and developing unique therapeutics targeted to specific tumorigenesis. This conceptual framework also provides insights into more complex tumor models in other organisms, including humans.

## Figures and Tables

**Figure 1 biology-13-00425-f001:**
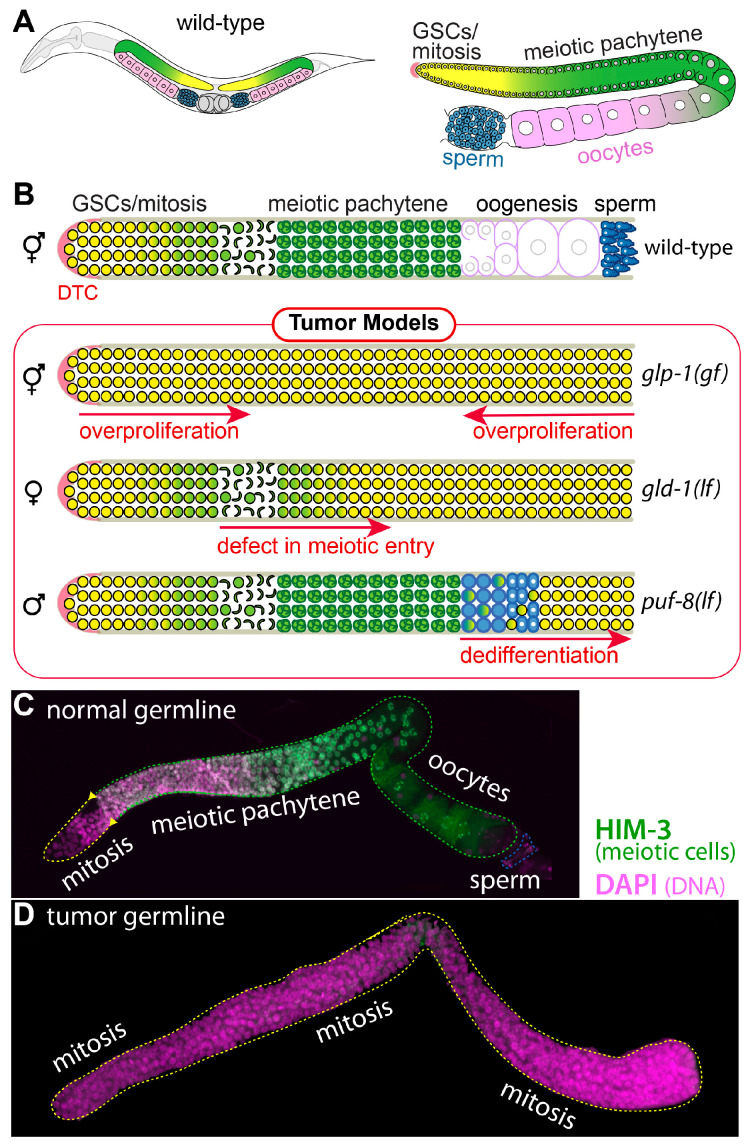
*C. elegans* germline and three distinct tumor models. (**A**) Schematics of adult *C. elegans* and its germline. Germ cells at the distal end of the germline, including GSCs, divide mitotically (yellow). As germ cells move proximally, they enter meiosis (green) and differentiate into either oocytes (pink) or sperm (blue). (**B**) Schematics of normal hermaphrodite germline and three tumor germline models resulting from *glp-1* gain-of-function (gf), *gld-1* loss-of-function (lf), or *puf-8* loss-of-function mutation (lf). (**C**,**D**) Normal and tumor germlines. Wild-type (N2) and tumor germlines were stained with anti-HIM-3 (meiosis marker) antibodies and DAPI (DNA marker).

**Figure 2 biology-13-00425-f002:**
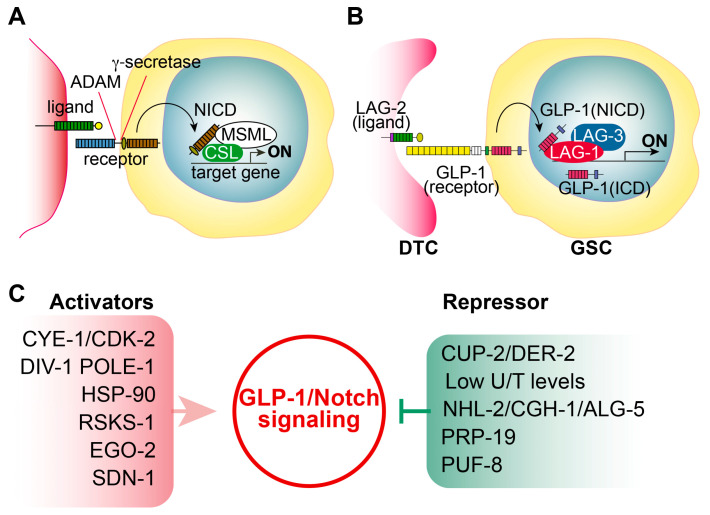
Notch signaling and its regulators. (**A**) Conserved Notch signaling pathways. Upon signaling, cleaved NICD translocates from the membrane to the nucleus. In the nucleus, NICD forms a tertiary complex with CSL and a co-activator (MSML, Mastermind-like protein), activating the expression of target genes. (**B**) *C. elegans* GLP-1/Notch signaling pathways. The DTC expresses GLP-1/Notch ligands (e.g., LAG-2) and employs GLP-1/Notch signaling to promote continued mitotic division of GSCs. (**C**) Positive and negative regulators of GLP-1/Notch signaling.

**Figure 3 biology-13-00425-f003:**
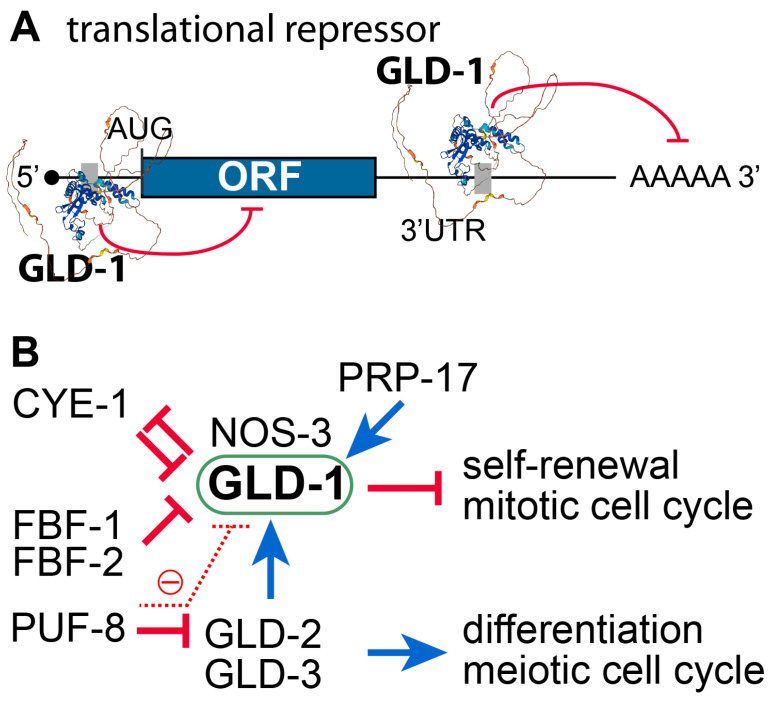
GLD-1 translational repressor and its regulators. (**A**) Schematic of GLD-1 binding to the regulatory element(s) (gray boxes) on the 5′ or 3′ UTRs of target mRNAs (a black line). GLD-1 generally represses the translation of target mRNAs. A predicted GLD-1 3D protein structure was generated using the AlphaFold Protein Structure Data Base [57]. (**B**) Positive and negative regulators of GLD-1.

**Figure 4 biology-13-00425-f004:**
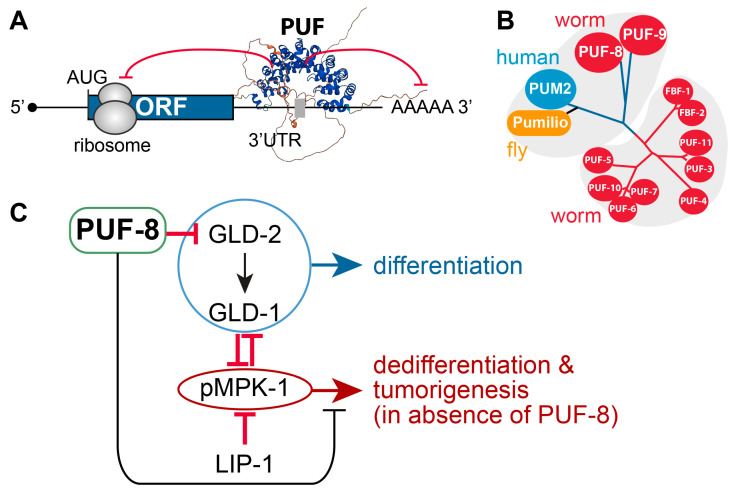
PUF-8 translational repressor and its regulators. (**A**) Schematic of PUF-8 binding to the regulatory element(s) (a gray box) on the 3′ UTR of target mRNAs (a black line). PUF-8 generally represses the translation of target mRNAs. A predicted PUF-8 3D protein structure was generated using the AlphaFold Protein Structure Data Base [57]. (**B**) The PUF protein family is widely distributed throughout eukaryotes. (**C**) Role of PUF-8 and its genetic partners in differentiation/dedifferentiation decisions.

**Table 1 biology-13-00425-t001:** The *glp-1* mutant alleles.

Allele	CGC Stock	Phenotype	Ref.
*bn18*	DG2389	Temperature-sensitive loss-of-function mutant	[12]
*q224*	JK1107	Temperature-sensitive loss-of-function mutant	[13]
*oz112*	-	A ligand-independent gain-of-function mutant characterized by the formation of germline tumors.	[14]
*ar202*	GC833	A temperature-sensitive gain-of-function mutant characterized by the formation of proximal (Pro) germline tumors. This phenotype differs from that of the *glp-1(oz112)* mutants. The *glp-1(ar202)* mutants develop “Pro” germline tumors due to delayed initial meiotic entry during the L4 stage at the restrictive temperature. However, our genetic results revealed that additional mechanisms may induce the formation of germline tumors, even in the adult stage	[15]

**Table 2 biology-13-00425-t002:** The *gld-1* mutant alleles.

Allele	CGC Stock	Phenotype	Ref.
*op236*	TG34	Fertile but hypersensitive to CEP-1/p53-mediated apoptosis	[59]
*q485*	JK3025	Sterile with germline tumors	[45]
*q268*	JK3025	Sterile with germline tumors	[45]
*q93*	JK3934	Sterile with germline tumors	[45]
*q343*	JK1058	Small abnormal oocytes	[45]

**Table 3 biology-13-00425-t003:** The *puf-8* mutant alleles.

Allele	CGC Stock	Phenotype	Ref.
*ok302*	JH1521	Fertile at 20 °C but sterile at 25 °C	[86]
*q725*	JK3231	Fertile at 20 °C, but some animals are sterile due to germline tumors at 25 °C	[87]

## Data Availability

Data are available for research purposes upon reasonable request to the corresponding author.

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
