# Peer review of "C. elegans Germline as Three Distinct Tumor Models"

_biology, 2024, doi:10.3390/biology13060425_

Round 1

Reviewer 1 Report

Comments and Suggestions for Authors

This manuscript provides an overview of three tumorigenic pathways that can produce germline tumors in C. elegans, along with lists of representative alleles and their regulators. The manuscript is well-written, clear and well-structured. I have a few suggestions that may improve the manuscript. First, it would be beneficial if the three tumor models were indicated early in the introduction or abstract. Currently, It is not clear what the three models are until after the introduction. As most of these genes involved in tumorigenesis are conserved, it would be informative to at least briefly mention relevant human cancer types for each tumor model, if available and applicable. Lastly, there appears to be a type in line 255: it references “four puf genes”, but five genes are listed in the following parentheses.

Reviewer 2 Report

Comments and Suggestions for Authors

The review describes three specific tumor model in th germline of C. elegans. Mutations in genes associated with the three different pathways lead to 1) aberrant proliferation 2) failure to differentiate 3) dedifferentiation of germ cells. The three different pathways and their specific regulators are described in details.

General comments

The three lists of proteins involved in signaling regulation are informative, however, they do not tell much about the molecular mechanisms of action of these proteins. For instance, Derlin proteins act in the ER and their impaired activity reduce GLP-1 dependent tumorigenesis. What could be the link ? What is the interpretation ? The three lists of regualtors are somewhat written in the same style. Genetic interactions are listed, but molecular mechanisms of regulation are not proposed. It is not obvious to always see how these different genetic factors could interact at the molecular level. I would advise to go carefull through the three lists of regulators and try to identify mechanisms of molecular functionning. This would greatly improve the quality of the review.

Specific comments:

line 47 "is a preferred genetic model for in vitro tumorigenesis research".

C. elegans is good model organism to study fundamental mechanisms of cell proliferation and differentiation and one possible model to study tumorigenesis in vivo (and not in vitro!). I would not say that it is a preferred model for tumorigenesis research, knowing that C. elegans only makes tumor in the germline.

line 162: fertile instead of fertility

line 168: what are the molecular nature of CYE-1 and CDK-2 ?

line 170: "proliferative fate of glp-1(bn18)". bn18 is the loss-of-function allele and does not show a proliferative fate. This is sentence is a bit confusing.

line 171: what does this sentence mean ? There is not enough information to understand it.

line 207: what is the molecular nature of the PUF-3 and PUF-11 proteins ?

line 208: what is ALG-5 ?

line 223: it would be interesting to know whether glp-1 a direct transcriptional target of AP-2?

paragraph 3.1: it would have been interesting to know if STAR proteins from other species are involved in germ line development.

line 241: human diseases.

Figure 3A: why is GLD-1 represented in the its 3D structure, since it is binding to a "line" representing the mRNA ? The folding of the protein is not mentioned in the text, neither referenced.

line 298: by "activating expression" you mean activating translation ? The term "expression" is used throughout the review in place of "translation". I find translation more precise compared to expression.

line 305: what do you mean by "inhibiting gld-2 mRNA" ?

line 306: the whole CYE-1/CDK2 section is very confused regarding the molecular mechanisms of GLD-1 regulation.

Figure 4A: why using the 3D structure of the PUF proteins ? What information does it bring to the picture ?

line 382: puf-8(lf) induces germline tumor in glp-1(ar202). It is not very clear knowing that glp-1(ar202) already exhibit tumorous germline

line 428: tumorigenesis pathway and line 431: tumour pathway. I am not sure to understand these two concepts. To my opinion, there are no pathways to tumors, it is the malfunctioning of pathways that are leading to tumorigenesis.
